# Transport for the Elderly: Activity Patterns, Mode Choices, and Spatiotemporal Constraints

**Yang Zhou** [1] **, Quan Yuan** [1] **and Chao Yang** [1,2,*]

1   Key Laboratory of Road and Traffic Engineering of the Ministry of Education, College of Transportation Engineering, Tongji University, Shanghai 201804, China; txzhpy@gmail.com (Y.Z.); quanyuan@tongji.edu.cn (Q.Y.)
2   Urban Mobility Institute, Tongji University, Shanghai 200092, China
*   Correspondence: tongjiyc@tongji.edu.cn; Tel.: +86-(021)-69-589-882

**Abstract:** The rapid aging of the population has posed significant challenges to society and raised new demand for transportation services. Understanding travel needs of the elderly is crucial to making effective strategies for accommodating their demand in many newly motorized cities in developing countries such as China. Using a Markov-chain-based mixture model, we identify two main activity patterns of the elderly: recreation-shopping-oriented (RS-oriented) pattern and schooling-drop-off/pick-up-oriented (SDP-oriented) pattern. Elderly people in the RS-oriented pattern enjoy a cozy life with much time spent on recreation and shopping activities, while those in the SDP-oriented pattern take responsibility of sending grandchildren to school and taking them back home. The RS-oriented elderly people are faced with spatial constraints to access the sparsely distributed recreational sights; however, the SDP-oriented group is subject to temporal constraints when making daily trips. These results would encourage policy makers to reconsider the role of transportation in aged people's lives and better accommodate their demand through designing safer walking and cycling environment and improving the quality of transit services.

**Keywords:** transportation policy; elderly; activity pattern; mixture model; clustering

---

## 1. Introduction

The aging of the population is a growing concern in many countries, including both developed and developing ones. For instance, the proportion of elderly people in the population has continuously increased in the last three decades, from 5.6% in 1990 to 11.9% in 2018, according to 2019 China Statistical Yearbook. It is predicted that this figure will reach 23.6% in 2050 [1]. The rapid aging process has increasingly attracted attention from the government and urged policy makers to reconsider the fundamental needs of this specific group of people in society. Elderly people can spend more time on shopping, recreation, and health-care needs, but have some disadvantages in mobility because of socioeconomic status and limited car ownership, in particular in developing countries [2]. Given the likely mismatch of supply and demand in daily travel of aged people, to understand their travel characteristics and needs could help policy makers provide better transport services in cities.

Travel behaviors and needs of the elderly in China are found to be different from many developed countries. Based on survey data in Nanjing, China, Feng (2017) found that elderly people are more inclined to go to vegetable markets than supermarket and do leisure activities in parks than gym/sports centers and museums [3]. Due to cultural differences, elderly people living with their adult children are likely to take the responsibility of childcare, such as sending children to school and taking them back home [4]. However, these characteristics have not been adequately studied in empirical studies.

In this paper, we apply a Markov-chain-based (MC-based) mixture model, which is widely used in pattern clustering, to categorize daily activity chains of the elderly. The clustering process results in two categories of patterns: recreation-shopping-oriented (RS-oriented) pattern and school-drop-off/pickup-oriented (SDP-oriented) pattern. The travel characteristics of these two patterns are quite different. Consequently, we raise some policy suggestions for two scenarios. The contributions lie in three main aspects: (a) we successfully separate the main activity patterns based on daily routine arrangement of travel among the elderly using an MC-based mixture model; (b) we attach importance to schooling drop-offs and pick-ups which are generally ignored by existing studies, and make comparative analyses with other activity patterns; (c) we make policy suggestions based on research findings to improve transportation services for the elderly.

The remainder of the article is structured as follows. Section 2 presents a literature review on travel behaviors of the elderly and patterns clustering techniques. Section 3 introduces methods used in the research, with a particular emphasis on the Markov-chain-based mixture model. Sections 4 and 5 focus on empirical results and discussions. The final section summarizes the findings and highlights future research directions.

## 2. Literature Review

### 2.1. Travel Needs and Behaviors of the Elderly

With much leisure time, the elderly could participate in more out-of-home activities and obtain socioeconomic opportunities [5,6]. Compared to other activities, they spend more time on shopping, family visits, recreation, and social activities [6]. Feng (2017) found that daily travel of the elderly is subject to the distribution of related facilities [3]. Their travel is highly relevant to vegetable markets, open space, and parks. In China, the unique household structure in which grandparents live together with their adult children has great effects on travel behaviors. Some of the travel needs of the elderly could be handled in joint family decision-making process [7]. For example, their adult children could drive to the supermarket and purchase essential goods for them. The elderly also take care of their grandchildren and arrange daily activities in relation to the kids' schedules.

Travel behaviors of the elderly are different from other age groups of people in quite a few aspects [8,9]. The elderly tend to travel less and make short trips in daily life compared with the younger population [10]. It is widely believed that vehicle availability and public transportation accessibility are the important factors that affect the mobility of the elderly. However, these effects are different across urban contexts. Elderly people travel in a car-oriented environment while seldom relying on public transit in developed countries including the United States [11,12], Canada [13], Australia [14], and the Netherlands [15]. Meanwhile, in many dense cities in developing countries, such as China, the elderly usually walk, ride traditional or electric bicycles, or use public transportation tools [9]. Given the differences in transport mode uses across contexts, studies have examined how the access to private vehicles can improve the quality of life among old people and indicated positive results [16,17]. When the elderly people can get access to more and better options, they would find travel easier and unrestricted, and make more trips [18,19]. For instance, bus passes can effectively stimulate extra trips of elderly people, allowing them to be more involved in their local community activities [20].

### 2.2. Activity Pattern Clustering Techniques

Clustering or grouping objects on the basis of their inherent similarity is an important technique in the field of pattern identification. Generally, there are two types of pattern clustering: hierarchical clustering and partitional clustering [21]. The former is to build a nested hierarchy of clusters by means of bottom-up merging or top-down separation, while the latter is a nonhierarchical clustering procedure that decomposes singletons into disjoint clusters. The partitional clustering includes distance-based and model-based types. The distance-based clustering model defines the intrinsic similarity between

pairs of observations via a variety of forms of distance, such as Minkowski distance, Pearson correlation, and Cosine similarity [22]. The model-based mixture model takes the cluster as a unimodal component within a mixture model, measuring similarity by probability distribution [23].

To cluster human activity patterns, scholars split an activity chain into discrete slices to construct a special sequence representing activity information. However, the similarity of these categorical sequences is difficult to measure in clustering. Mohamed et al. (2016) divided 1-week smart card data into 168 hourly slices and applied a mixture of unigrams model to cluster profiles based on the number of trips within each slice [24]. Joh et al. (2001) handled the similarity among categorical vectors with the weighted sum of identical attributes [25]. Jiang et al. (2012) [26] and Goulet-Langlois et al. (2016) [27] converted the slice sequence into a flatted binary vector to calculate distance. Others also applied string or sequence clustering algorithms. For instance, Wang et al. (2018) [28] and Zhai et al. (2019) [29] measured categorical sequence distance with an edit distance which is the essential number of substitutions, insertions and deletions when converting a string into a target one. However, the edit distance did not consider the characteristics of each change operation.

Given the advantages and disadvantages of different approaches, we, in this article, analyze travel behavior of the elderly by clustering activity patterns using a Markov-chain-based mixture model. This model is a combination of the Markov chain model and mixture model, which are suitable to handle categorical time series sequence. It captures the temporal characteristics in two aspects: transitions between different activities representing sequence order, and transitions of same activities keeping the information of duration. The transition distribution regarding the similarity between pairs of sequences makes more sense in characterizing activity temporal information than the traditional binary vector or edit distance.

## 3. Materials and Methods

### 3.1. Data

Considering the request of accuracy in activity time and type, we use household travel survey data to identify activity patterns. The data were collected in Nanjing across six discrete weekdays in 2015. In total, 30,780 households and 69,904 respondents were interviewed in the survey and 16% were elderly people aged 60 and over. Each respondent was asked to report their socioeconomic characteristics and recall details of daily travel during the surveyed day. The recorded details included origins and destinations, travel time, transport mode used, trip purpose, and land use category of home community, etc. Travel diary of an individual contains a sequence of trips, in each of which he/she travels from the previous stop to the next one for a specific purpose. According to the nature of trip making, we define an activity as the gap between two consecutive trips. Subsequently, we link all trips in order by inserting activities one by one and eventually get the activity chain (see Figure 1a,b).

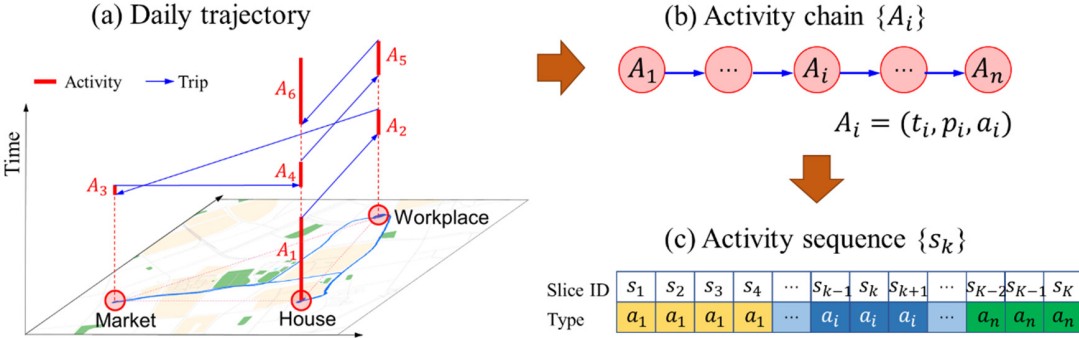

**Figure 1.** Process of activity chain segmentation. (**a**) Daily trajectory; (**b**) activity chain; (**c**) activity sequence.

## 3.2. Activity Chain Segmentation

A daily activity chain of an individual is a set of sequential activities, written as $\{A_i\}$, ($1 \leq i \leq n$). Each activity is a vector composed of time, position, and type, denoting $A_i = (t_i, p_i, a_i)$, where activity type $a_i \in \mathbb{Z}$. The $\mathbb{Z}$ is a set of all possible activity types which are derived from a variety of trip purpose types originally in the household travel survey (see Table 1). In this study, $\mathbb{Z} = \{H, W, S, E, B, D, R, O\}$; each letter represents Home, Work, Shopping, Education, Business, Drop-off/pickup, Recreation, Others, respectively.

**Table 1.** Trip purpose coding from the original survey data.

| No. | Symbols | Combined Categories | Original Categories in Survey |
|:---:|:---:|:---:|:---:|
| 1 | H | Home | Go home (owned or leased) |
| 2 | W | Work | Go to work<br>Return to work |
| 3 | S | Shopping | Buy goods (groceries, clothes, appliances)<br>Buy meals |
| 4 | E | Education and training | Attend school as a student<br>Attend training or course class |
| 5 | B | Business/obligations | Attend business meeting<br>Agricultural/fishing events<br>Use professional services<br>Other business |
| 6 | D | Drop-off/pickup | Drop-off/pickup someone<br>Accompany kids to school/training class |
| 7 | R | Recreational/social | Recreational activities<br>Social and community<br>Visit public places<br>Visit friends, relatives, patients<br>Go for a walk |
| 8 | O | Others | Something else |

Regarding the structure of activity chains, it would be challenging to efficiently compare the chains between different individuals due to the variances in the length of each activity. Therefore, we divide time of a day into minimal time slots to develop activity sequences in a uniform format. Activity sequences can be denoted as $\{s_k\}$, ($1 \leq k \leq K$), in which $K$ is the length of the sequence (see Figure 1c). The activity category of each time slot is assigned as follows: $s_k = a_i$ if more than half of the time within $s_k$ is occupied by the activity $A_i$. The overall process of activity chain segmentation is shown in Figure 1. In our study, we select the minimal time slot as 30 min, resulting in 48 slots within a day (i.e., $K = 48$). These activity sequences will be the input in the clustering model to identify daily activity patterns.

## 3.3. Clustering Using MC-Based Mixture Model

To understand the overall picture of daily activity patterns among a large number of individuals is highly challenging, despite the efforts of defining activity chains using uniform time slots. We determine to cluster all the daily activity chains into categories and focus on major categories of patterns in the empirical analysis. Different from the traditional distance-based clustering algorithms which measure similarity with distance, model-based mixture models create each cluster as a unimodal component [23]. Particularly, the Markov-chain-based mixture model is based on a collection of finite probability distributions underlying the Markov chain [30]. Similar to other mixture models, it assumes that each cluster follows a probability distribution, and the mixture density (also called clustering kernel) is expressed as the sum of densities of all the components, as given by

$$f(\mathbf{y}|\vartheta) = \sum_{g=1}^{G} \pi_g f_g(\mathbf{y}|\vartheta_g) \tag{1}$$

where $G$ is the total number of components; $\pi_g$ is the $g$-th mixing proportion, $\sum_{g=1}^{G} \pi_g = 1$ and $\pi_g > 0$ for all $g$ from 1 to $G$; $f_g(y|\vartheta_g)$ is the $g$-th component density, and all of them follow the same type of distribution, $\vartheta_g$ denotes the corresponding parameters; $\vartheta$ is a vector of collective parameters, $\vartheta = (\pi_1, \ldots, \pi_G, \vartheta_1, \ldots, \vartheta_G)$.

For activity sequence clustering, distance measure problem of categorical variables is well-solved by the mixture model, but how to characterize the time series turns out to be another challenge. Here, we apply the Markov chain to the component density function. The Markov chain is a stochastic process with the Markov property that the probability of an observation in any state is influenced only by the observations of a small number of immediately preceding states. The first-order Markov process is a particular situation that the observation at time $s$ is dependent on its preceding observation at time $s - 1$, i.e., $P(y_s|y_{s-1}, \ldots, y_1) = P(y_s|y_{s-1})$. Given the activity sequence of an individual $u$, the first-order Markov chain model can be characterized by an initial probability $\alpha$ and a transition matrix $\xi_{pq}$, expressed, respectively, as $\alpha = P(y_{u1} = p)$ and $\xi_{pq} = P(y_{us} = q|y_{u,s-1} = p)$, where $p$ and $q$ belong to the state category set $\mathbb{Z}$. The probability of the $u$-th sequence can be calculated by

$$P(\mathbf{y} = \mathbf{y}_u) = P(y_{u1}) \prod_{s=2}^{S} P(y_{us}|y_{u,s-1}) = \prod_{p=1}^{C} \alpha_p^{I(y_{u1}=p)} \cdot \prod_{p=1}^{C} \prod_{q=1}^{C} \xi_{pq}^{N_{pq}} \tag{2}$$

where $S$ is the length of sequence; $C$ is the number of state categories, $C = 8$ for $\mathbb{Z}$; $I(\cdot)$ is the indicator function; $N_{pq}$ is the frequency of transitions from state $p$ to state $q$ in the sequence.

Then, the Markov chain model is put into the component density $\{f_g(y|\vartheta_g)\}$, and the mixture density can be rewritten as

$$P(\mathbf{y}|\vartheta) = \sum_{g=1}^{G} \pi_g \prod_{p=1}^{C} \alpha_{g,p}^{I(y_{u1}=p)} \cdot \prod_{p=1}^{C} \prod_{q=1}^{C} \xi_{g,pq}^{N_{pq}} \tag{3}$$

where $\alpha_{g,p}$ and $\xi_{g,pq}$ are the Markov parameters that belong to cluster $g$.

To estimate parameters $\vartheta = (\pi_g, \alpha_{g,p}, \xi_{g,pq})$, the maximum likelihood estimation carried out by means of the Expectation-Maximization (EM) algorithm was applied [31,32]. For all the old people $U$, the likelihood function can be constructed as

$$P(\mathbf{y}|\vartheta) = \prod_{u=1}^{U} \sum_{g=1}^{G} \pi_g P(y_u|\vartheta_g) = \prod_{u=1}^{U} \sum_{g=1}^{G} \pi_g \prod_{p=1}^{C} \alpha_{g,p}^{I(y_{u1}=p)} \cdot \prod_{p=1}^{C} \prod_{q=1}^{C} \xi_{g,pq}^{N_{pq}} \tag{4}$$

Assume that $\mathbf{z}_{ug}$ represents the probability that activity sequence of the $u$-th individual belongs to cluster $g$. In the E-step, we calculate the posterior probability $\mathbf{z}_{ug}$ based on the raw sequence data under the initial parameters of mixture model. In the M-step, we calculate the model parameters for each cluster based on sequences with the allocated cluster. At the $m$-th iteration, the updates of posterior probability $\mathbf{z}_{ug}^{(m)}$ in E-step and model parameters $\vartheta^{(m)}$ in M-step contribute to the increase in the observed log-likelihood. The $\mathbf{z}_{ug}^{(m)}$ is expressed in Equation (5), and $\vartheta^{(m)}$ is calculated in Equations (6)–(8). The iteration ends when the probability $\mathbf{z}_{ug}$ reaches the maximum value.

$$\mathbf{z}_{ug}^{(m)} = \frac{\pi_g^{(m-1)} \prod_{p=1}^{C} \left(\alpha_{g,p}^{(m-1)}\right)^{I(y_{u1}=p)} \prod_{p=1}^{C} \prod_{q=1}^{C} \left(\xi_{g,pq}^{(m-1)}\right)^{N_{u,pq}}}{\sum_{g'=1}^{G} \pi_{g'}^{(m-1)} \prod_{p=1}^{C} \left(\alpha_{g',p}^{(m-1)}\right)^{I(y_{u1}=p)} \prod_{p=1}^{C} \prod_{q=1}^{C} \left(\xi_{g',pq}^{(m-1)}\right)^{N_{u,pq}}} \tag{5}$$

$$\pi_g^{(m)} = \frac{1}{U} \sum_{u=1}^{U} \mathbf{z}_{ug}^{(m)} \tag{6}$$

$$\alpha_{g,p}^{(m)} = \frac{\sum\limits_{u=1}^{U} \mathbf{z}_{ug}^{(m)} I(y_{u1} = p)}{\sum\limits_{u=1}^{U} \mathbf{z}_{ug}^{(m)}} \tag{7}$$

$$\xi_{g,pq}^{(m)} = \frac{\sum\limits_{u=1}^{U} \mathbf{z}_{ug}^{(m)} x_{u,pq}}{\sum\limits_{u=1}^{U} \mathbf{z}_{ug}^{(m)} \sum\limits_{q'=1}^{C} x_{u,pq'}} \tag{8}$$

We use the Bayesian information criterion (BIC) to determine the number of clusters [33]. The BIC is expressed as

$$BIC = -2\sum_{u=1}^{U} \log P(\mathbf{y}|\hat{\vartheta}) + N \log U \tag{9}$$

where $\log P(y|\hat{\vartheta})$ is the log-likelihood; $N$ is the total number of parameters, and $N = G \cdot C^2 - 1$. The clustering performs best when the BIC gets the minimum.

## 4. Results

### 4.1. Categorization of Activity Patterns

Initially, we use the MC-based mixture model to cluster activity patterns from a two-dimensional matrix which has 69,904 residents in the row and 48 time series slices in the column. The number of clusters is determined as three, based on the BIC index. In addition, in this scenario, the patterns exhibit significant differences from each other, as shown in Figure 2. Cluster #1 in general displays a working and education-oriented lifestyle, cluster #2 has strong reliance on shopping and entertainment activities, while cluster #3 is most affected by drop-off or pickup activities. These drop-off/pickup activities are likely to be school drop-off/pickup (SDP) because 90% of their destinations are located in education-related land uses, according to our further analysis. Therefore, activity patterns of residents can be classified into working–education-oriented (WE-oriented), recreation-shopping-oriented (RS-oriented), and school-drop-off/pick-up-oriented (SDP-oriented), in terms of daily activity scheduling.

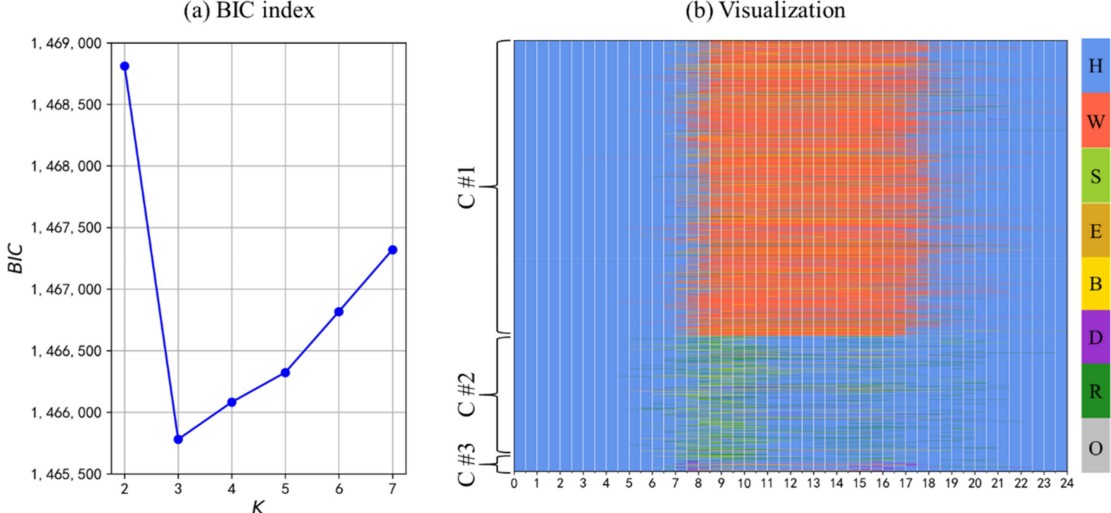

**Figure 2.** Activity patterns clustering for all residents. (**a**) Parameter determined by the Bayesian information criterion (BIC) index; (**b**) clustering results visualization. Notes: H = Home, W = Work, S = Shopping, E = Education, B = Business, D = Drop-off/pick-up, R = Recreation, O = Other.

For the elderly, the great majority of them fall into the RS-oriented and SDP-oriented patterns, which is expected given their social roles and behaviors. We extract data of the age over 60 and use the MC-based mixture model for clustering. As shown in Figure 3, there are 10,737 old people pertaining to the RS-oriented pattern and 427 pertaining to the SDP-oriented pattern. In spite of a small share, the SDP-oriented pattern represents a typical lifestyle of the elderly. With the rapid process of urbanization in China, more and more old people have been changing their family role from the retired idlers to child-guardians and taking the responsibility of school drop-offs and pick-ups to reduce the burden of breadwinners. To some extent, they play a semi-commute role in daily life because they are restricted to the routine schedules of schooling while not linked to workplaces. Overall speaking, to better understand the way that elderly people make transport choices and chain their daily activities would be rather helpful for improving policy making and providing better transport services for those with limited options in daily travel.

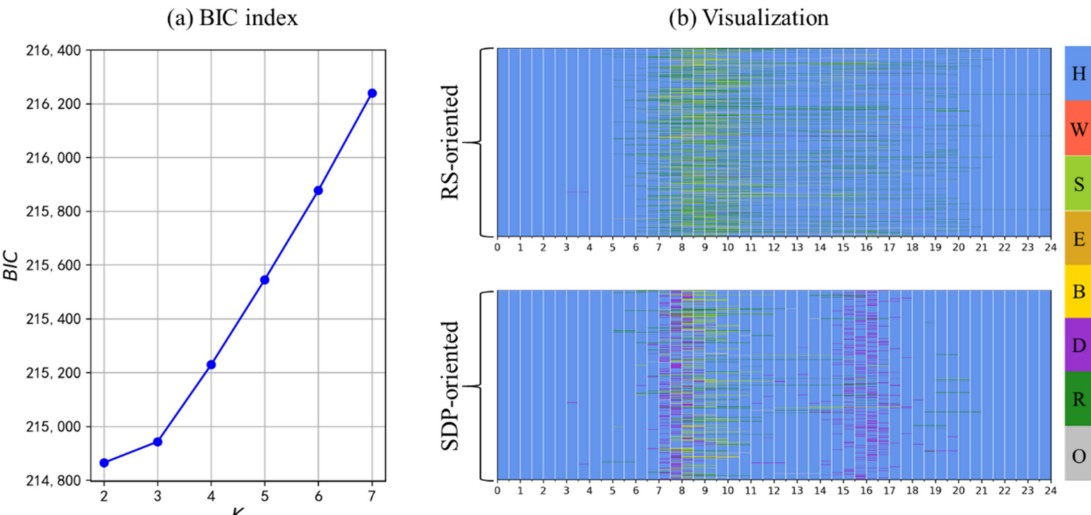

**Figure 3.** Activity pattern clustering for the elderly. (**a**) Parameter determined by the BIC index; (**b**) clustering results visualization. Notes: H = Home, W = Work, S = Shopping, E = Education, B = Business, D = Drop-off/pick-up, R = Recreation, O = Other.

### 4.2. Summary of Activity Patterns Across Two Categories

Through activity pattern clustering and summarizing, we were able to identify major patterns in the two categories mentioned above (shown in Figure 4). From the perspective of daily activity chaining, aged people in the RS-oriented category are primarily involved in two simple patterns—"Home-Recreation-Home" ("H-R-H", 37%) and "Home-Shopping-Home" ("H-S-H", 36%). By contrast, those in the SDP-oriented category have more complex patterns of activity chains related to SDP activity.

Figure 5 presents the characteristics of the occurring time of activities in two categories. Each panel stands for the cumulative frequency distribution of activities at each time slot. In the RS-oriented category, people go shopping and do recreational activities in morning peak hours from 7 a.m. to 11 a.m. and afternoon peak hours from 2 p.m. to 4 p.m. The morning peak is significantly more packed than the afternoon peak. The recreational activities are relatively expansive during the day; starting from 6:30 a.m. until 5 p.m., such activities could account for at least 10% of the time. Recreational activities and shopping activities are also similar in terms of occurring time.

In the SDP-oriented category, more than 20% of grandpas and grandmas drop off children between 7 a.m. and 8 a.m. and pick them up between 3:30 p.m. and 4 p.m. Compared to the earlier category, the peaks during the day appear to be shorter and highly restricted to schooling schedules. The elderly people belonging to this category spend very limited time on activities other than SDP, especially after the a.m. peak hours.

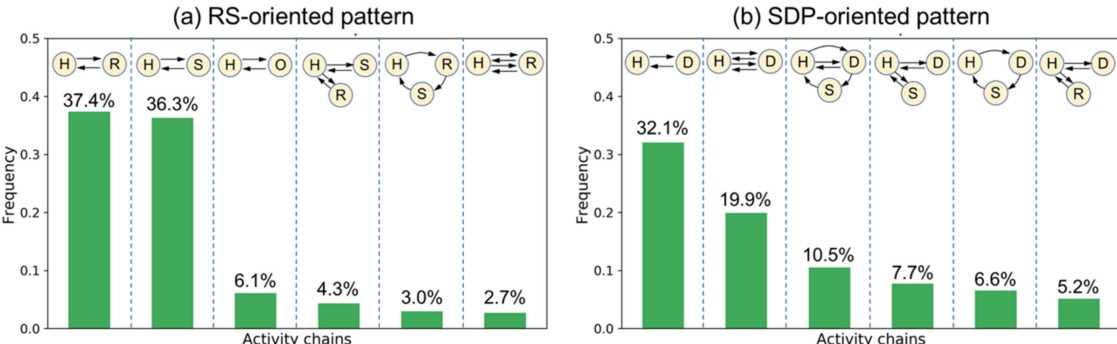

**Figure 4.** Fraction of dominant activity chains in the elderly's patterns. (**a**) Recreation-shopping (RS)-oriented pattern; (**b**) school-drop-off/pick-up (SDP)-oriented pattern. Notes: H = Home, R = Recreation, S = Shopping, D = Drop-off/pick-up, O = Other.

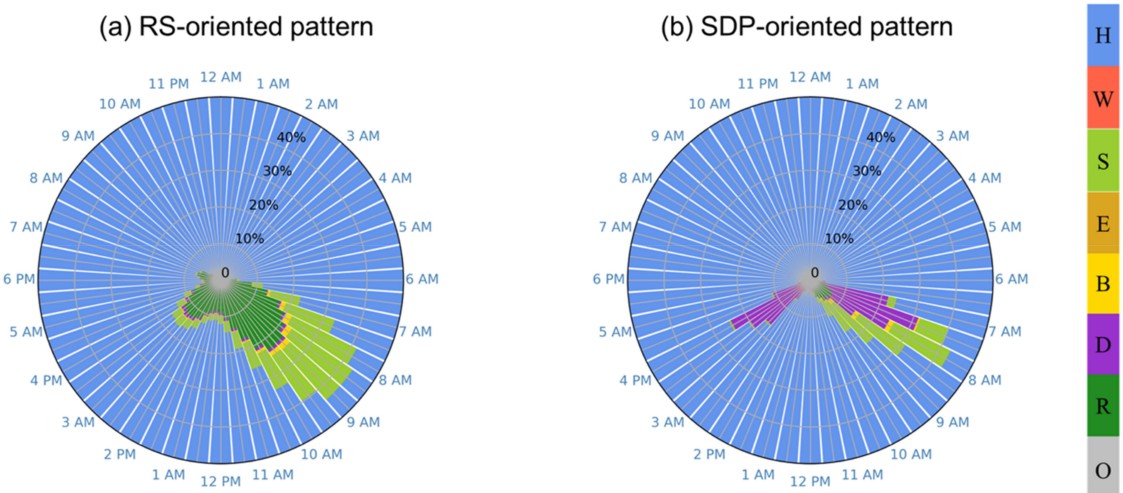

**Figure 5.** Activities occurring at each time of day. (**a**) RS-oriented pattern; (**b**) SDP-oriented pattern. Notes: H = Home, W = Work, S = Shopping, E = Education, B = Business, D = Drop-off/pickup, R = Recreation, O = Other.

### 4.3. Transport Mode Choices

As discussed in the literature review, travel mode choices are the outcomes of tradeoffs in family decision-making process, and highly related to personal characteristics, vehicle ownership, travel distance, and trip purpose, etc. However, elderly people, compared to the younger adults, as a whole, are faced with disadvantages of physical and financial capabilities. Such disadvantages are, in particular, significant in developing Asian countries, where younger family members are usually prioritized in using private vehicles to commute. Therefore, many Chinese cities have provided affordable public transport services to help the disadvantaged group through offering discounted or free transit passes, reducing the boarding and alighting difficulties, and installing priority seats in vehicles.

Figures 6 and 7 show that adults aged 18 to 59 years and the elderly aged over 60 greatly differ in travel distance and transport mode uses. Younger adults travel an average of 5.3 km in their commuting trips, relative to no more than 3 km in other types of trips. Among the three types of trips the two groups share—recreation trips, shopping trips, and schooling drop-off and pick-up trips—younger adults apparently travel longer distance and use more private vehicles. In either of the three types of trips, elderly people rely heavily on nonmotorized modes.

For the elderly, the majority of them access their shopping and SDP destinations within a radius of 2 km away from home, while doing recreational activities in a larger geographic area. They usually

only take responsibility for schooling drop-offs and pick-ups near home, leaving long-distance SDP to younger adults who drive more often. In addition, they can do shopping and purchase daily essentials in close-by general stores. However, recreational sites favored by the elderly such as scenic areas and natural parks are in many cases not as evenly distributed as shopping and schooling opportunities. Those recreational sites are too sparsely located that aged people cannot access by foot or bicycle; instead, they have to rely on public transit including bus and subway.

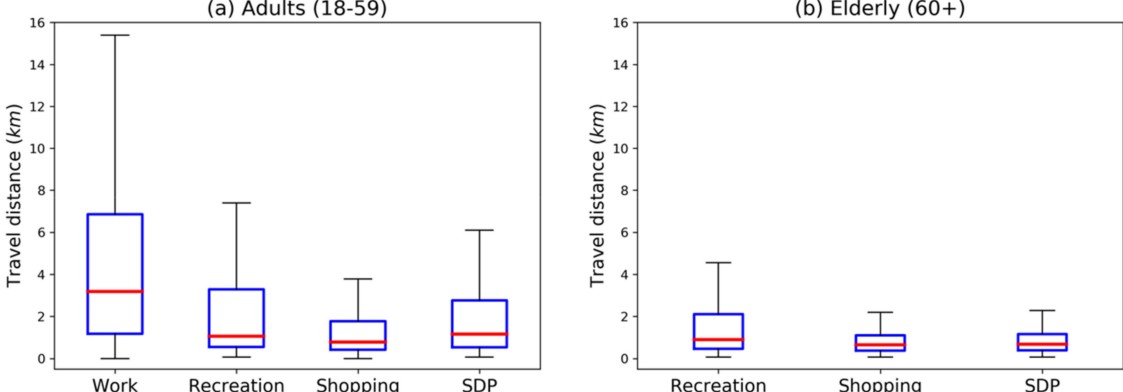

**Figure 6.** Distribution of travel distance. (**a**) Travel distance of adults aged 18 to 59 years; (**b**) travel distance of the elderly aged 60 and over. Notes: five important lines from bottom to top in boxplot represent the minimum (excluding outliers), first quartile, medium, third quartile, maximum (excluding outliers) value, respectively.

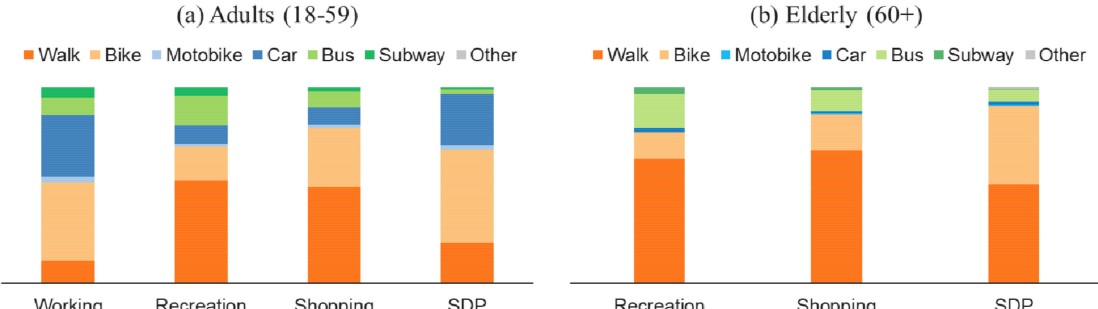

**Figure 7.** Shares of transport mode choices. (**a**) Adults aged 18 to 59 years; (**b**) elderly aged 60 and over.

Therefore, the elderly people are faced with considerable constraints in both recreation and SDP-oriented travel. The unavailability of nearby recreation sites leads to long-distance travel, causing a spatial mismatch of travel demand and supply. On the other hand, schooling drop-offs and pick-ups require stringent time windows which are usually during peak hours. Making such trips in a crowded environment can be challenging, especially for aged adults. The temporal constraints could eventually contribute to potential conflicts and safety threats.

## 5. Discussion

### 5.1. School Drop-Off/Pick-Up Safety

In China, young parents living in cities attach great importance to their child's education from primary school to high school due to the "one-child" policy and social and cultural traditions. They may choose to rent a housing unit near the school (especially located in the city center) to allow more time accompanying and supervising their children. Elderly people, as grandparents, usually drop off and pick up children to reduce the burden of working adults. However, drop-off and pick-up times set by schools are embedded during peak hours, so elderly people with those responsibilities are exposed to traffic safety threats when roads are messily congested. Safety issues of schooling pick-ups have

been heatedly discussed on the Internet in China. The main factors of the elderly getting injured when dropping off and picking up their grandchildren include traffic disorder during peak hours at the school gate, weak awareness of road safety, and careless use of bikes or e-bikes. The SDP-oriented pattern identified above suggests that policy makers should pay more attention to the traffic safety concern among elderly grandparents during peak hours (in particular around 7 a.m. and 4 p.m. on weekdays). To carefully examine the traffic conditions in close vicinity of schools and evaluate how the traffic safety of the elderly grandparents is threatened during peak hours can be a good option for policy makers who would like to address the concern.

*5.2. Bus Services for Recreational Activities*

Recreational activities are very valuable to the elderly people who seek peaceful and relaxed lifestyles. Outdoor ones are particularly attractive; old people enjoy going for a walk nearby home, visiting natural parks, and doing exercise in the open air. They are willing to travel for a long distance to access those sites and facilities. Transit services have become the most reliable choices in such travel, due to the constraints of transport mode options.

The quality of transit services differs across places. High-density transit networks have been developed in the urban area, but many suburban and rural areas are still underserved. Figure 8 shows locations of recreation destinations as well as bus services among both elderly visitors and their younger counterparts in two suburban counties in Nanjing. While those suburban areas contain quite a few attractive outdoor reservoirs, parks, and scenic areas, it is relatively difficult for elderly people to access those public recreational resources without private vehicles. Once the resources are located away from bus transit lines, very few old visitors had left their footprints there while more younger visitors could make it to access those resources probably by car.

In this sense, it is necessary for transport planners to reconsider the bus services connecting suburban recreational sites from the perspective of elderly visitors. Improved bus services can greatly help accommodate the travel demand of those visitors with limited transport mode choices. For instance, short-distance bus lines connecting county center to the parks and reservoirs would be helpful. Such lines, compared to long-distance lines originating from urban areas, can be more reliable and affordable.

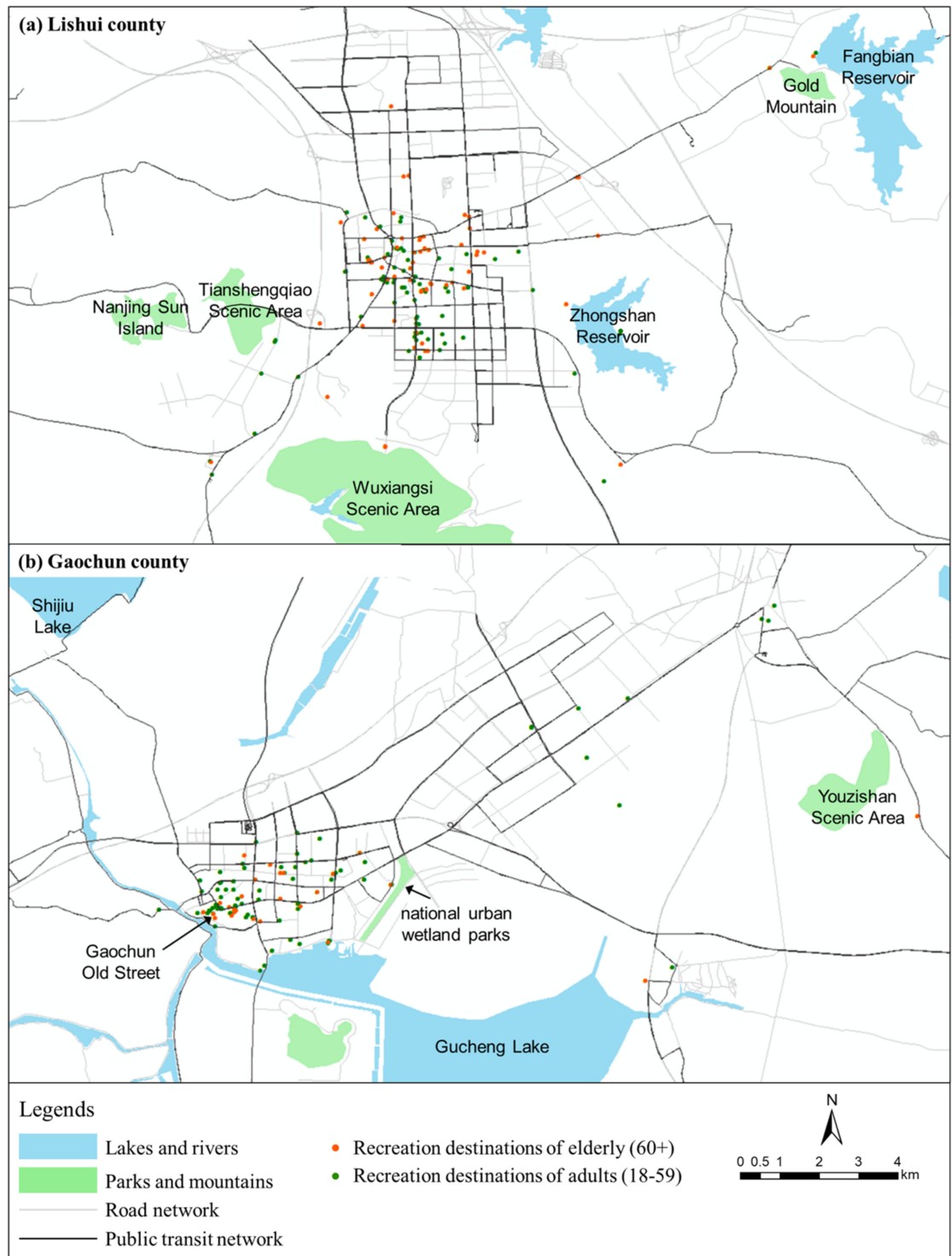

**Figure 8.** Activity locations and transit conditions in different regions. (**a**) Lishui county; (**b**) Gaochun county.

## 6. Conclusions

Based on Nanjing household travel survey data, we identified two main activity patterns of the elderly using the Markov-chain-based mixture model. One is the recreation-shopping (RS)-oriented pattern, and another is the school-drop-off/pickup (SDP)-oriented pattern. From the clustering results, recreation chains and shopping chains are similar to each other, but both are quite different from SDP-oriented chains. While old people in the SDP-oriented pattern have some temporal restrictions in

daily activity arrangement, those in the RS-oriented pattern have relatively more flexibility in time scheduling but are constrained by limited destination options across space.

The elderly travel farther away to access recreational resources than shopping. They could meet daily shopping needs by going to stores nearby but may have no choice but taking buses to reach faraway recreation-oriented destinations. Many elderly people drop off their grandsons and granddaughters at school near home and pick them up by foot or bicycle. During the peak hours, they are faced with a high risk of getting injured on the congested roads. These characteristics of travel among the elderly people can effectively help policy makers manage the citywide transport system to better accommodate the needs of this group of people.

This work still goes with some inherent limitations that can be studied in the future. The dataset we used is travel data covering trips of each person interviewed on a survey day. Therefore, we cannot access the travel data of his/her family members who were absent at the interview time, nor can we get the respondents' travel behaviors in the long term. The activity pattern clustering approach can help produce more interesting results with multiple-day travel survey data.

**Author Contributions:** The authors confirm contribution to the paper as follows: conceptualization, Y.Z., Q.Y. and C.Y.; methodology, Y.Z. and Q.Y.; software, Y.Z.; validation, Y.Z., Q.Y. and C.Y.; formal analysis, Y.Z.; data curation, Y.Z.; writing—original draft preparation, Y.Z. and Q.Y.; writing—review and editing, Y.Z. and Q.Y.; visualization, Y.Z.; supervision, Q.Y. and C.Y. All authors have read and agreed to the published version of the manuscript.

**Funding:** This research received no external funding.

**Acknowledgments:** The authors sincerely acknowledge the support from Nanjing government for providing the household travel survey dataset.

**Conflicts of Interest:** The authors declare no conflict of interest.

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
