# Peer review of "Transport for the Elderly: Activity Patterns, Mode Choices, and Spatiotemporal Constraints"

_sustainability, doi:10.3390/su122310024_

Round 1

Reviewer 1 Report

By applying a Markov Chain-based (MC-based) mixture model, this study tried to categorize daily activity chains of the current elderly people in China. And the author(s) succeed to find that there are two categories of patterns: i.e. recreation-shopping-oriented (RS-oriented) pattern and school-drop-off/pickup-oriented (SDP-oriented) pattern.

The research questions in this study are quite clear and the methods are appropriate. A table and figures are helpful to understand the point of results and discussions. Also, because this study focuses on the lifestyles of current Chinese senior citizens, the findings in this study have academic novelty. 

In addition, this kind of studies will be applicable to Chinese real policies such as supporting elder people, universal design, and city planning.

Author Response

Thanks a lot for your comments.

Reviewer 2 Report

The article deals with a very important and very current issue, which sees the elderly as a resource of high value for the sustainable progress of society.

The classification algorithm proposed by the authors and applied to the study of a sample of travel diaries in China, made it possible to classify the activities of the elderly and to characterize them in relation to their space-time impact.

The analysis of a significant sample group has made it possible to identify two groups of the elderly population with different daily activities and habits, and to compare them with those of the remaining adult population.

The characterization of the two clusters has made it possible to make extremely useful considerations on the activities / habits of the elderly population, which are extremely interesting for planning urban and mobility policies. With this knowledge, these policies can be effectively calibrated taking into account the needs and requirements of protection and security of an ever-growing segment of the population, with invaluable contributions to the development of the whole society.

I would like to thank the authors for their nice contribution, which I consider for the publication as is.

Author Response

Thank you so much for recognizing the contributions of this paper.

Reviewer 3 Report

This is a very well documented research in the field. English should be checked and updated accordingly. 

Activity patterns are mostly related to socioeconomic status, car ownership, trip purpose, land uses etc HOWEVER miss the key issue in terms of the urban planning environment, the existence of soft mobility infrastructure (sidewalks, narrow streets, cycling lanes, parklets... etc.), the availability of mixed land uses, the existence of small neighbourhood parks and many more. Transport mode choices are the result of the urban environment (attractive urban environment, security, lighting, greenery...) and the available choices in the field (walking, cycling facilities), along with the parameters discussed in lit. review and section 4.3. 

There is adequate analysis for the sequence of trips/ activities and the method/ models used. However the paper shall update the analysis scope in terms of what is the real problem in mode choices and identify the current benefits of the current patterns, i.e. the fact that the elderly walk within a 2km radius away from home is in line with the current trends in mobility plans - see 15min city plans in Paris.

The discussion shall also be updated including proposals for new recreational spaces in the neighbourhood scale, new bus services related to the current and future walking/ cycling routes and everything else that the authors consider important within the current mobility trends. 

Author Response

Thank you for your critical comments.

(1) We strongly agree with your views that travel mode is highly related to the mobility infrastructure and environment. In this paper, we focus on the impacts of trip purpose and public transit conditions on travel mode choices of the elderly, and simplify the problem in our discussions. If we have multiple sources of data, we may do it in our future research.

(2) Generally speaking, the elderly tend to travel less and make short trips in the daily life. While in China, parts of the elderly are also willing to cost much time on their way to the scenic areas for sightseeing. The elderly are more inclined to do leisure activities in parks than gym/sports centers and museums (Feng et al., 2017). Based on our data, we find the similar phenomena that lots of people aged over 60 took public transports to the parks and scenic areas. Apart from the daily life range (15min city plans), the accessibility to those heated seeing sights is also important, even though these places may be located far away from communities.

(3) Thank you for your suggestion. This paper is trying to recognize activity patterns of the elderly and discuss their travel issues and disadvantages for two scenarios. The recreation-shopping-oriented (RS-oriented) elderly are faced with spatial constraints to access the sparsely distributed recreational sights, however, the schooling-drop-off/pick-up-oriented (SDP-oriented) group is subject to temporal constraints when making daily trips. The topics you mentioned are what we want to do in future. The proposal of policies will need more thorough research.

Reference:

  1. Feng, J. (2017). The influence of built environment on travel behavior of the elderly in urban China. Transportation Research Part D: Transport and Environment, 52, 619-633.